



# Mixed-phase Direct Numerical Simulation: Ice Growth in Cloud-Top Generating Cells

Sisi Chen[1], Lulin Xue[1], Sarah Tessendorf[1], Kyoko Ikeda[1], Courtney Weeks[1], Roy Rasmussen[1], Melvin
Kunkel[2], Derek Blestrud[2], Shaun Parkinson[2], Melinda Meadows[2], and Nick Dawson[2]

[1]National Center for Atmospheric Research (NCAR), Boulder, CO.
[2]Idaho Power Company, Boise, ID.

*Correspondence to*: Sisi Chen (sisichen@ucar.edu)

**Abstract.** A detailed microphysical model is developed using a Lagrangian-particle-based direct numerical simulation
framework to simulate ice growth in a turbulent mixed-phase environment. The Lagrangian particle method is employed to
track the interactions between ice, droplets, and turbulence at the native scales. The investigation reveals for the first time the
mixed-phase processes at the sub-meter length scales using direct numerical simulation.

This paper examines the conditions that favor effective ice growth in the cloud top generating cells. Investigations over a
range of environmental (macrophysical and turbulent) and microphysical conditions (ice number concentrations) that
distinguish generating cells from their surrounding cloudy air were conducted. Results show that high liquid water content
(LWC) or high relative humidity (RH) is critical to maintaining effective ice growth and mixed-phase. As a result,
generating cells with high LWC and high RH provide favorable conditions for rapid ice growth. Sensitivity studies on ice
number concentrations show that when the ice number concentration is below $1\ cm^{-3}$, a typical range in the mixed-phase
clouds, a high LWC is needed for efficient formation of big ice particles. The study also found that supersaturation
fluctuations due to small-scale turbulent mixing have a negligible effect on the particle mean radius but substantially broaden
the size spectra which can affect the subsequent collection process.

## 1 Introduction

It is well-recognized that the model representation of natural ice is not accurate due to a poor understanding and/or
representation of microphysical processes such as the ice nucleation mechanism, representation of supercooled liquid water,
and the interactions between the ice-liquid phase at sub-grid scales (Korolev and Milbrandt 2022; Korolev et al. 2017; Koop
and Mahowald 2013; Ovchinnikov et al. 2014). This poor representation is exacerbated by the insufficient measurements to
fully constrain the microphysical parameterization at sub-grid scales (Fridlind et al. 2012; Baumgardner et al. 2017;
Heymsfield et al. 2017; Korolev et al. 2017; Field et al. 2017). This uncertainty in the growth of ice can further impact the
forecast of the supercooled liquid water amount in mixed-phase clouds (e.g., Morrison and Pinto 2006; Barrett et al. 2017)
via the Weber-Bergeron-Findeisen (WBF) mechanism (i.e., ice grows at the expense of liquid evaporation as a result of the
difference in saturation vapor pressures with respect to liquid and ice, e.g., Storelvmo and Tan 2015) and thus the subsequent
precipitation process.

The interactions between the ice particles and the supercooled liquid water in a turbulent environment is a multi-scale issue,
and detailed information, such as the Lagrangian history of the particle growth, the spatial distribution of the particles of
distinct phases at fine scales are key information to the decay and maintenance of mixed-phase clouds (Korolev and
Milbrandt 2022). Theoretical, modeling, and observational studies have been conducted to examine the influence of various
factors on mixed-phase clouds such as the entrainment-mixing, WBF process, large eddies, vertical oscillations, and ice
number concentrations (Hoffmann 2020; Korolev 2007, 2008; Korolev and Field 2008; Khain et al. 2022; Morrison et al.
2012; Lee et al. 2021). Nevertheless, the native scales at which interactions between individual ice particles, liquid droplets,



and turbulence were rarely explored due to the limitation of the spatial resolution of the models and in-situ measurements. Lagrangian-particle-based direct numerical simulations (DNS) can fill this knowledge gap in fine-scale microphysics and dynamics by resolving the turbulence and cloud particle growth.

DNS has been applied in cloud microphysics modeling since the early 2000s (Vaillancourt et al. 2001, 2002) to study the turbulent impact on particle growth and spectral broadening. A majority of the research (e.g., focused on the warm-phase clouds only (Ayala et al. 2008; Grabowski and Wang 2013; Li et al. 2020; Gotoh et al. 2016, 2021; Saito et al. 2019; Chen et al. 2018a, b, 2020, 2021); only a few studies have touched the ice-phase clouds (Vowinckel et al. 2019a, b) while no attention has been given to the mixed-phase processes.


In this study, a Lagrangian-particle-based mixed-phase microphysics was implemented in the DNS model to untangle the interactions between ice, droplets, and turbulence at true native scales relevant to microphysics and to improve the fundamental understanding of ice growth in mixed-phase clouds. This paper aims to present a mixed-phase DNS model with the capability of simulating the microphysical processes key to the development, maintenance, and decay of mixed-phase
environments and the evolution of the ice and droplet size distributions.

Specifically, this study investigates the influence of cloud top generating cells (GCs) on ice growth in mixed-phase clouds. GCs are fine-scale structures commonly observed in different types of clouds such as the winter orographic clouds (Tessendorf et al. 2019; Kumjian et al. 2014), midlatitude cyclones (Keeler et al. 2016; Rauber et al. 2015), and mixed-phase
clouds in the Southern Ocean (Wang et al. 2020). The GCs typically contain more liquid and ice and stronger updrafts than the adjacent portion of the clouds. Therefore, it is a key region for the formation of ice and precipitation. To resolve processes in GCs, an accurate representation of microphysics and dynamics are needed. The processes involve interactions between ice particles and liquid particles at centimeter scales. The structure of the flow at this length-scales is difficult to measure using in-situ data and is also unable to resolve using the WRF type mesoscale model. Therefore, a mixed-phase
DNS is beneficial to obtain a process-level understanding at such fine scales.

With the overarching research question of "how can we use this innovative tool to design experiments and improve ice parameterizations in models that do not resolve mixed-phase microphysics explicitly?", our model development efforts aim to:


1) Present a DNS model studying the mixed-phase cloud microphysics with resolved fine-scale turbulence and Lagrangian cloud particles (aerosols, droplets, and ice crystals) and demonstrate the capability of such a model.

2) Investigate the impact of turbulence and environmental conditions on the ice growth and cloud glaciation at the native scales of droplet-ice interactions. We explored the potential mechanism of efficient ice growth in GCs.
Controlled experiments are conducted by the mixed-phase DNS to quantitatively examine the impact of the GCs environment on the ice growth. In-situ measurement data from the Seeded and Natural Orographic Wintertime clouds - the Idaho Experiment (SNOWIE) (Tessendorf et al. 2019) are used to set up the initial conditions for the microphysics and environmental conditions which resemble those inside and outside GCs. Finally, the atmospheric implications of the simulations will be discussed.

**2 Mixed-phase DNS model**

This study is part of our ongoing model development of a detailed cloud microphysics model Chen et al. (2016, 2018a, 2018b, 2020, 2021). In this paper, the first version of the mixed-phase DNS model is introduced to tackle the interactions between droplets, ice, dynamics, and thermodynamics at microscales. The modeling framework of the warm-phase DNS by



Chen et al. (2018b) was adopted. The following two model improvements were made to consider the ice depositional growth
in a turbulent-mixing environment:

(1) added depositional growth of spherical ice particles, and

(2) added the external forcing on the thermodynamic fluctuation fields (i.e., temperature, $T'$, and water vapor mixing ratio, $Q_v'$) to account for the effects of turbulent mixing cascading down from larger scales.

## 2.1 Depositional growth of ice particles

The purpose of this model is to simulate the growth of small ice particles ($r < 100 \, \mu m$) at the early stage of the mixed-phase

clouds to examines the impacts of microphysical/environmental conditions on the production of larger ice particles (i.e., $r \geq$

$100 \, \mu m$) and to understand the Wegener-Bergeron-Findeisen (WBF) process (Pruppacher and Klett 2010)(Pruppacher & Klett

2010), which refers to the rapid growth of ice at the expense of the evaporation of droplets when supercooled droplets and ice

crystals co-exist, but also impacted by fine-scale turbulence in clouds. In this study, the deposition growth of newly nucleated

ice crystals was considered in the model as the only mechanism for ice growth. Other nucleation and growth mechanisms such

as the freezing of supercooled droplets, secondary ice productions, ice aggregation, and riming were not considered in this

study. The ice depositional growth rate in mass of each individual spherical ice crystal was calculated in the model (Rogers

and Yau 1989, equation (9.4)):

$$\frac{dm}{dt} = \frac{4\pi C S_i}{\left(\frac{L_s}{R_v T} - 1\right)\frac{L_s}{K'T} + \frac{RT}{e_{sat,i}D'}} \qquad (1)$$

where $m = \frac{4}{3}\pi r^3 \rho_i$ is the mass of a spherical ice crystal with a radius of r, $C = r$ is the capacitance for spherical ice crystal,

$S_i$ is the supersaturation over the ice surface, and $L_s = 2.83 \times 10^6 \, J \, kg^{-1}$ is the latent heat required to deposit per unit mass

of water vapor to ice. $R_v = 467$ is individual gas constant for water vapor, $K'$ and $D'$ are, respectively, the thermal conductivity

and water vapor diffusivity that include kinetic effects (see equation 11a-b in Grabowski et al. 2011 for the detailed

expressions), and $e_{sat,i}$ is the saturation water vapor pressure. The assumption of a circular ice particle may not accurately

represent the true geometry of observed ice particles whose true habit depends on temperature and size (Korolev and Isaac

2003). Model development is undergoing to consider ice habits in future simulations.

## 2.2 Thermodynamic forcings on fluctuation fields

To account for the effects of turbulent mixing between different parts of the cloud cascaded down from scales larger than DNS

domain size (i.e., $L \gg 1 \, m$), a forcing scheme was implemented to modulate the fluctuations of the temperature field ($T'$) and

the water vapor mixing ratio field ($Q_v'$). In this way, statistical stationarity of thermodynamic fluctuations imposed by large-

scale turbulence is achieved:

$$\frac{\partial T'}{\partial t} = -\nabla \cdot (UT') - W'\Gamma_d + \frac{L}{c_p}C_d' + \frac{L_i}{c_p}C_i' + D_t\nabla^2 T' + F_T \, , \qquad (2)$$

$$\frac{\partial q_v'}{\partial t} = -\nabla \cdot (Uq_v') - (C_d' + C_i') + D_v\nabla^2 q_v' + F_{q_v} \, , \qquad (3)$$

In these equations $U$ is the flow velocity, $W'$ is the vertical perturbation velocity, $C_d'$ and $C_i'$ are the differential droplet

condensational rate and differential ice depositional rate, respectively, between the grid cell and the entire domain. $D_t$ and $D_v$



are thermal diffusivity and water vapor diffusivity. $F_T$ and $F_{q_v}$ are forcing terms. The two fluctuation fields were forced in a low-wavenumber band in the Fourier space, similar to the way the velocity field is forced (Chen et al. 2016). The energy cascades down to smaller and smaller scales via the non-linear interactions between eddies is eventually dissipated by viscosity

and thermal diffusivity at millimeter scales. The forcing intensity, which measures the strength or efficiency of the turbulent mixing of the scalar fields ($T$ and $q_v$), is determined by the prescribed standard deviation ($\sigma T, \sigma q_v$) which can be obtained from flight measurements:

$$F_T = C_{T'}\delta T \tag{4}$$
$$F_{q_v} = C_{q_v'}\delta q_v \tag{5}$$

Here $C_{T'}$ and $C_{q_v'}$ are constants determined by the domain size of the simulation.

## 3 Experimental setups

To set up the model initial conditions, in-situ measurements during the intensive observation period (IOP) 22 of the Seeded and Natural Orographic Wintertime clouds - the Idaho Experiment (SNOWIE, see Tessendorf et al. 2019) was used. IOP22

was observed from 2017-03-09 14:00-18:00 UTC. We utilized the data from flight segment "A" to extract the environmental conditions of the generating cells. In segment A (14:08-14:10 UTC, with flight distance of ~ 12 km given a mean aircraft speed of 98.9 m/s), 13 GCs in total were identified. Six of the GCs had at least 5 seconds of flight measurements, which translates to a horizontal length scale of roughly 500 m or longer. We selected the time series of GC that lasted the longest in the measurement period (i.e., the largest generating cell of the five) as the statistics were believed to be more representative of GC

conditions than the rest of the timeseries. Table 1 summarizes the environmental conditions measured in this segment. As demonstrated, the observed number concentrations of drizzle and large ice particles (d $> 100\ \mu m$) in the GC environment are much higher than that of non-GC environments.

**Table 1.** The environmental and microphysical conditions of the GCs from SNOWIE IOP22's flight segment A.

| Variable | Domain average value |
|---|---|
| Mean pressure (P) | 571.6 hPa |
| Mean temperature (T) | 259.53 K |
| Mean mixing ratio ($Q_v$) | $1.66 - 2.17\ g\ kg^{-1}$ |
| Standard deviation of temperature ($\sigma T$) | 0.143 K |
| Standard deviation of water vapor mixing ratio ($\sigma Q_v$) | $0.045\ g\ kg^{-1}$ |
| Total drizzle number concentration | $0.4 - 1\ L^{-1}$ (GCs); 0.05-0.25 $L^{-1}$ (non-GC) |





| Total ice number concentration (d > 100 $\mu m$) | 0.57– 0.9 $L^{-1}$ (GCs); 0.23 – 0.51 (non-GC) |
|---|---|


Two sensors (Rosemount and DMT100) were available to measure the water vapor mixing ratio which has uncertainty in the mean values (1.66 $g\ kg^{-1}$ from DMT100 and 2.17 $g\ kg^{-1}$ from Rosemount), therefore, both values are tested to account for the measurement uncertainty. In this study, the Bergeron process is focused on, in which the water vapor is supersaturated over ice but subsaturated over a liquid surface so that ice grows at the expense of droplet evaporation. Therefore, a diagnostic

relation between the supersaturation and temperature was calculated using a simple steady-state parcel model to identify the range of relative humidity (RH) favoring the Bergeron process (as shown in the triangular shaded area in Figure 1). To ensure that  (1) the tested range covers both conditions that are favorable or unfavorable for the Bergeron process, and (2) the corresponding mixing ratio at this condition reflects the measured range and its uncertainty (Table 1), we chose a range in supersaturation between $S_w = -20\% \sim 5\%$, corresponding to RH = 80~105% (purple stripe in Figure 1) at the given

temperature and pressure condition of the GCs was chosen as the initial conditions in the DNS. The microphysical response to varying moisture levels was therefore investigated.

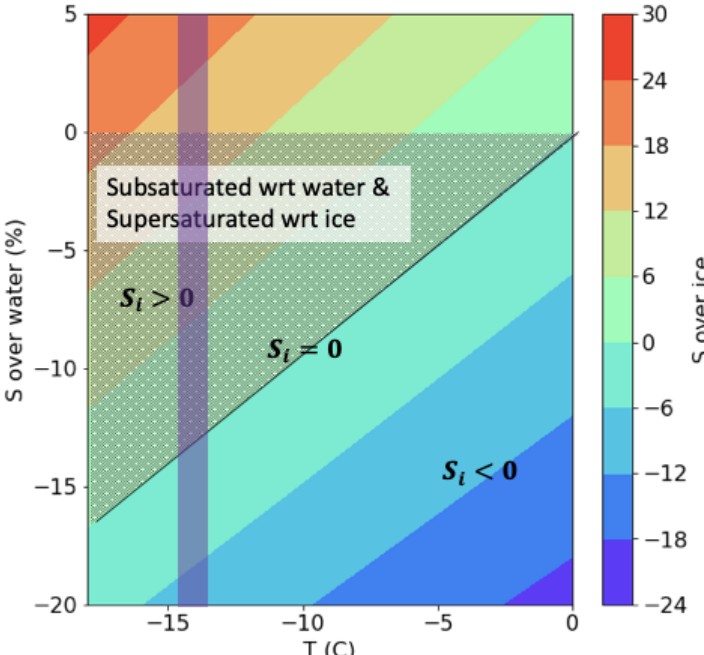

**Figure 1.** Diagnostic relationship of the temperature (x-axis), supersaturation over the liquid water (y-axis), and

supersaturation over ice (color contours). The shaded gray triangular area indicates the condition in favor of the Bergeron process, and the purple column marks the mean temperature of the generating cell measured in IOP22 (T~ -13.6 C). The range of supersaturation (-20%-5%, i.e., RH=80-105%) in the y-axis will be examined in the current study.

Six groups of experiments, including the control group, were performed, which consist of 18 simulations in total (see summary

of the experiments in Table 2). Within each group, only one condition is varied and the rest remains the same. The initial conditions for the controlled experiment are listed in Table 2. Droplets have a dry radius of 1 μm with a hygroscopicity parameter $\kappa = 0.3$. The purpose of conducting these sensitivity tests is firstly to validate the model on whether it could produce




reasonable results, and secondly, by tuning different controlling variables, to understand what conditions favor ice growth and explore why the generating cell environment tends to produce more ice.


**Table 2.** Summary of the initial conditions of the DNS experiments conducted in this study. In each experimental group, only the conditions that are different from the Control Group are listed in the remaining groups.

| Experiment Group | | Initial conditions |
|---|---|---|
| 1 | Control | Relative humidity (RH) = 90% <br> Ice concentration ($N_i$) = $1\ cm^{-3}$ <br> Ice initial radius ($R_i$) = $1\ \mu m$ <br> Droplet concentration ($N_d$) = $100\ cm^{-3}$ <br> Droplet initial radius ($R_{wet}$) = $10\ \mu m$ <br> Droplet dry radius ($R_{dry}$) = $1\ \mu m$ <br> Hygroscopicity parameter $\kappa = 0.3$ <br> Liquid Water Content (LWC) = $0.55g\ m^{-3}$ <br> Mean updraft velocity $\bar{w} = 0\ ms^{-1}$ <br> Eddy dissipation rate $\epsilon = 10m^2s^{-3}$ <br> Domain size $V = 20\ cm\ \times 20\ cm\ \times 20\ cm$ |
| 2 | RH_GC | RH = 80, 85, 90, 95, 100, 105% |
| 3 | RH_noGC | RH =80, 85, 90, 95, 100, 105%, <br> $R_{wet}$ = 1.5 μm, <br> LWC = $0.002\ g\ m^{-3}$ |
| 4 | High_IN | $N_i$ = $1, 10, 100\ cm^{-3}$ |
| 5 | Low_IN | $Ni = 0.008, 0.08, 0.8\ cm^{-3}$ <br> $LWC = 0.055g\ cm^{-3}$ <br> $N_d = 10\ cm^{-3}$ |
| 5 | Turb | With/without turbulence ( $\epsilon = 0, 10m^2s^{-3}$) |

The GCs are reported to contain a higher liquid water content (LWC) than the surrounding cloud body. Therefore, two groups
of experiments were conducted, which highlight a high LWC and a low LWC condition, respectively. The high LWC experiments were initialized with LWC = $0.55\ g\ m^{-3}(R_{wet} = 10\ \mu m, N_d = 100\ cm^{-3})$, a typical value observed in a GC during SNOWIE, and therefore are named as RH_GC. Likewise, the low LWC experiments were initiated with LWC = 0.002



$g\ m^{-3}(R_{wet} = 1.5\ \mu m, N_d\ =\ 100\ cm^{-3})$, a much lower value than inside the generating cell, and are named RH_noGC (see Table 2 for setup detail).


Another two groups of experiments were conducted with different initial ice number concentrations. The first group is named High_IN in which the small ice number concentrations vary from 1 to 100 $cm^{-3}$ with the same initial radius of 1 $\mu m$ (Table 2). The second group is named Low_IN with $N_i$= 8, 80, 800 $L^{-1}$. This range is commonly observed in many natural mixed-phase clouds.


To examine how turbulent-mixing-induced supersaturation fluctuations affect the ice growth and the glaciation process, the last group, named Group Turb, compares simulated results with and without turbulence. The model configurations are the same as the control group except that the experiment without turbulence has no fluctuations in the velocity, temperature, and mixing ratio fields. In other words, the model is treated as a traditional cloud parcel with no external forcing term and turbulent
transportation term in the temperature and mixing ratio equations (Equation (2-3)) to generate supersaturation fluctuations.
.

## 4 Results

### 4.1 Macrophysical impact

We first investigate the impacts of initial macrophysical conditions (i.e., mean quantities) on the ice growth. We emphasize
the relative humidity and liquid water content, as the two are directly related to the water availability and immediately affect the ice growth.

Figure 2 shows the response of mean particle radius and mean supersaturation to different initial RH. In RH_GC group, the mean state of supersaturation quickly adjusts to the steady-state within 25 seconds (Figure 2 (c-d)). All air parcels become
nearly saturated with respect to water and supersaturated over ice ($S_i \approx 10\%$) and remain statistically steady thereafter. In fact, the steady-state saturation ratio is maintained by an unstable microphysical condition under which the Bergeron process (the ice grows rapidly at the expense of droplet evaporation) is active (Figure 2 (a-b)). The loss of water vapor due to the ice growth is continuously replenished by the evaporation of droplets, leading to a steady RH.

The RH_GC experiments demonstrate that the saturation at its steady-state is insensitive to the initial RH with abundant supercooled liquid water (high LWC). During the steady state, all ice grows at a relatively constant rate and is less sensitive to the initial relative humidity compared to droplet growth. All air parcels tend to be glaciated, and the rate of glaciation depends on the initial RH. As Figure 3 demonstrates, a low RH parcel glaciates faster than that of a high RH but produces a lower ice water content (IWC) due to less available total water vapor. In the atmosphere, when patches of droplets or drizzle drops are
falling through or transported to a sub-saturated environment from elsewhere through processes such as strong turbulent-mixing and fall streak of hydrometeor (drizzle or melted snow) from the GCs, they can quickly replenish the surrounding air and create favorable conditions for ice growth. In the meanwhile, the cloud region with high RH environment, such as in the GCs due to its more upward motion than the surrounding environment, can maintain mixed-phase longer, but also generate a high IWC due to the high availability of water vapor and liquid water.


In RH_noGC environments, due to insufficient liquid water in the parcel, ice growth rate is highly dependent on RH (Figure 2e). For RH <90%, the air parcel is too dry to maintain ice growth due to low availability in both liquid water and water vapor, reaching a negative steady-state supersaturation (Figure 2g). For RH>=90%, the air parcel is still supersaturated with respect to ice (Figure 2g) and thus positive ice growth is maintained. The ice grows at a much slower rate compared to its counterpart
in the RH_GC group. It is shown that the air parcel generally glaciates faster regardless of its lower IWC (Figure 3c, d).





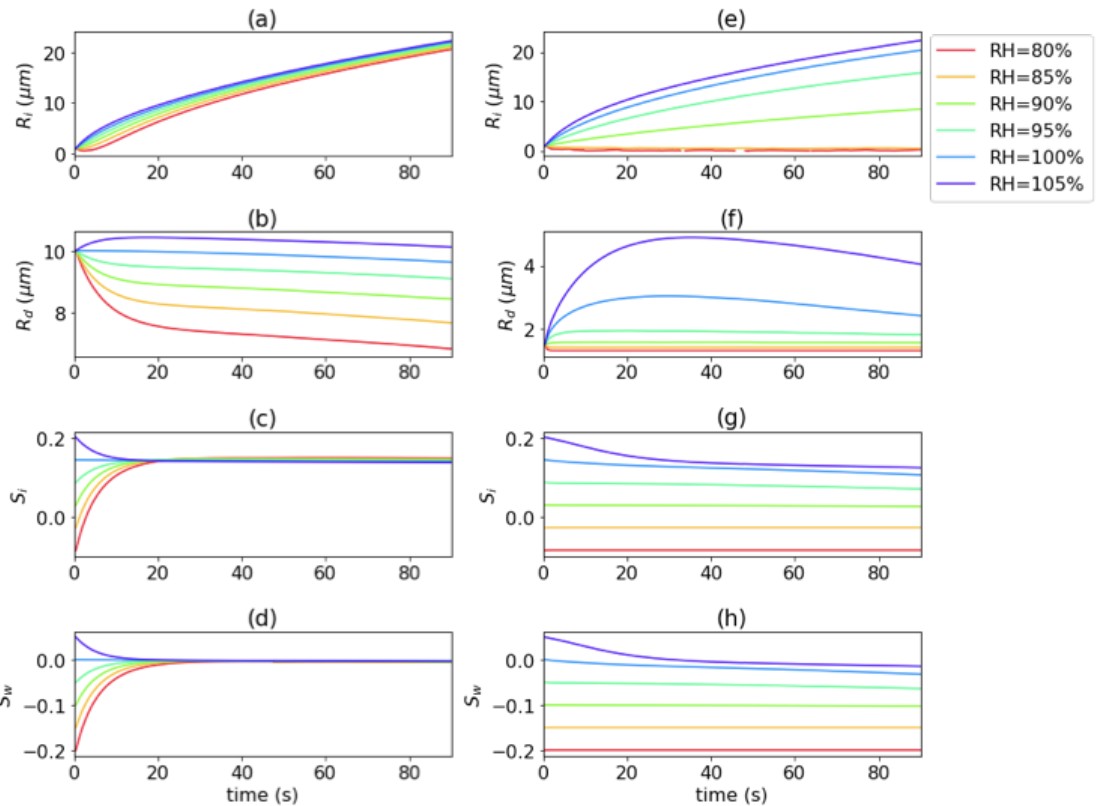

**Figure 2.** From top to bottom: the evolution of mean ice radius, mean droplet radius, mean supersaturation over ice, and mean supersaturation over water responding to different initial RH. The left panel demonstrates results from a high LWC case (initial LWC= 0.55g m$^{-3}$, Group RH_GC), and the right panel from a low LWC case (initial LWC= 0.002g m$^{-3}$, Group RH_noGC)

The impact of RH on the glaciation rate is also prominent. Compared to the low RH cases, the air parcel initialized with a high RH glaciates slower but produces ice water faster (Figure 3). This is because a high RH can delay and slow down the Bergeron process. It is shown that maintaining a high RH is important for efficient ice growth while maintaining mixed-phase for a longer period. In RH_GC group, droplets with RH=105% maintain growth for the first few seconds (Figure 2b); in RH_noGC, droplets maintain high growth rates in the first 30 seconds (Figure 2f). This difference in the two groups is caused by the difference in the initial droplet wet radius. RH_noGC was initiated with droplets with wet radius of 1.5 $\mu m$ and RH_GC with 10 $\mu m$, while droplets in both groups have the same dry radius of 1 $\mu m$. Following the $\kappa$-Kohler theory (Petters and Kreidenweis 2007), the effective saturation ratio that considers the solute and curvature effects at the droplet surface is $S_{sat} = \frac{R_{wet}{}^3 - R_{dry}{}^3}{R_{wet}{}^3 - R_{dry}{}^3(1-\kappa)} exp(\frac{2\alpha_w}{R_v \rho_w T R_{wet}})$ (see equation (1) in Chen et al. 2020 for the detailed explanation of the terms and constants), where $\kappa$ is the hygroscopicity parameter, $G_{solu} = \frac{R_{wet}{}^3 - R_{dry}{}^3}{R_{wet}{}^3 - R_{dry}{}^3(1-\kappa)}$ is the solute term, and $G_{curv} = exp(\frac{2\alpha_w}{R_v \rho_w T R_{wet}})$ is the curvature term. $G_{curv} \approx 1.0$ in our case, and $S_{sat} \approx G_{solu}$. The droplet growth rate is proportional to $S - S_{sat}$, where S is the environmental saturation ratio. Therefore, either a large S or a small $S_{sat}$ can lead to an effective droplet growth.





In RH_noGC, the droplets have a large dry radius a thin wet layer to begin with ($G_{solute} \approx 0.89 << 1.0$ with $R_{wet} = 1.5\,\mu m$ and $R_{dry} = 1\,\mu m$ ), the solute term has an immediate strong impact on the growth rate in cases with RH≥ 100%, i.e., $S - S_{sat} >> 0$. One can observe the prominent droplet growth in Figure 2f at the beginning, which is absent in RH_GC case (Figure 2b). When droplets grow to a larger size, $R_{wet}$, the growth rate slows down due to a higher effective saturation ratio and a lower environmental saturation ratio (i.e., a smaller $S - S_{sat}$). For example, for a droplet grows to $R_{wet} = 5\,\mu m$, $S_{sat} \approx G_{solu} \approx 0.998$. With the environmental saturation ratio becoming subsaturated at a later time ($S - S_{sat} < 0$), a negative growth rate is expected. In RH_GC, the solute effect is not important ($G_{solu} \approx 1.0\ for\ R_{wet} = 10\,\mu m$) and therefore, the growth rate is mainly determined by the environmental saturation ratio.

In summary, the value of supercooled liquid imposes an opposite impact on the ice growth and glaciation rate: a higher LWC can generate a higher IWC but results in a slower glaciation rate. To maintain both effective ice growth and mixed-phase for a long time, the environment needs either a high RH or a source of liquid water. Generating cells with a higher LWC and RH than the adjacent portion of the clouds provide ideal environments for efficient ice growth. In the meantime, solute effects from large aerosols can be important at the initial growth stage to maintain droplet growth and thus the increased LWC; this has an important implication for future cloud seeding operations to a maintain high LWC for a more active Bergeron process.

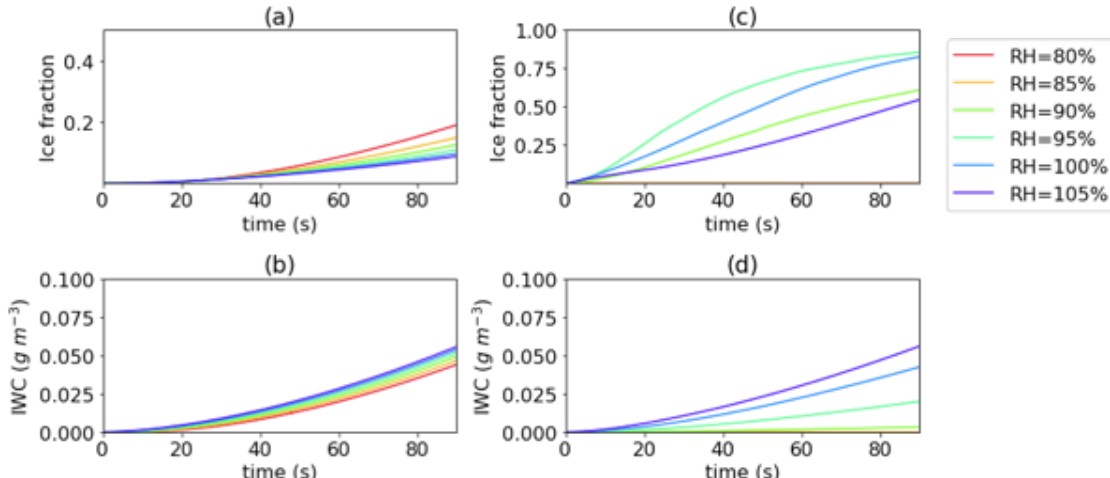

**Figure 3.** (a-b) The evolution of ice fraction and ice water content, respectively, responding to different initial RH with an initial LWC of 0.55 g m$^{-3}$. (c-d) same as (a-b) but with an initial LWC of 0.02 g m$^{-3}$. The ice fraction is defined as the ratio between ice water content and total water content.

We also examined the evolution of particle size distribution (PSD) and found that it is highly sensitive to RH and LWC available in the air parcel. In RH_noGC, the RH plays a critical role in determining the mean radius: the higher the RH, the larger the ice mean size (Figure 4(b)). At RH<95%, the majority of droplets evaporate and ice growth is inhibited.





**Figure 4.** From columns (a-d): the evolution of the (a) droplet size distribution and (b) ice size distribution in RH_noGC; and the evolution of (c) droplet size distribution and (d) ice size distribution in RH_GC. From top to bottom, the initial relative humidity increases from 80% to 105%. Colors indicate the number concentration of the individual size bin.


Given that all ice particles have the same initial size, the broadening of the spectral width is solely due to the turbulent fluctuation of the supersaturation field. The width of the ice spectral peaked at RH = 90% where the supersaturation with respect to ice is close to 100% (Figure 2g). Ice particles can experience positive or negative supersaturation in their immediate vicinity as they are advected by the turbulent flow. This results in a highly variable Lagrangian growth history within the ice

population and thus the widest spectra. For cases with RH < 90% where $S_i \ll 0$ when reaching steady-state, the growth of the





entire ice population is inhibited; for cases with RH>90% where $S_i \gg 0$ at steady-state, the effect of mean supersaturation dominates the fluctuations, and ice growth is less diversified. This can also be seen in RH_GC where $S_i \gg 0$ in all simulations, the impact of initial RH on the mean ice radius and spectral width is negligible, and the ice PSDs are very similar (Figure 4(d)).

The effective impacts of supersaturation fluctuations on spectral widening can be quantified by the relative dispersion, which is a measure of the relative width of the PSD and is defined as the standard deviation of the particle radius normalized by its mean radius. In RH_GC, the relative dispersion of the ice size spectra only differs initially during the RH adjusting to the steady-state, and converge to the same width at the end (Figure 5a). A mildly negative relation between the initial RH and relative dispersion of droplet size spectra is found (Figure 5b), indicating that supersaturation fluctuations, which broadens the

size spectra, play a more important role in a low sub-saturated environment. In RH_noGC, the spectral broadening for both ice and droplets is more sensitive to the initial RH. Different from RH_GC, the relative dispersion of droplet size spectra shows a reverse trend and increases with RH (Figure 5d). This reverse trend is because most of the droplets evaporate in the low RHs and therefore reach a nearly-zero relative dispersion.

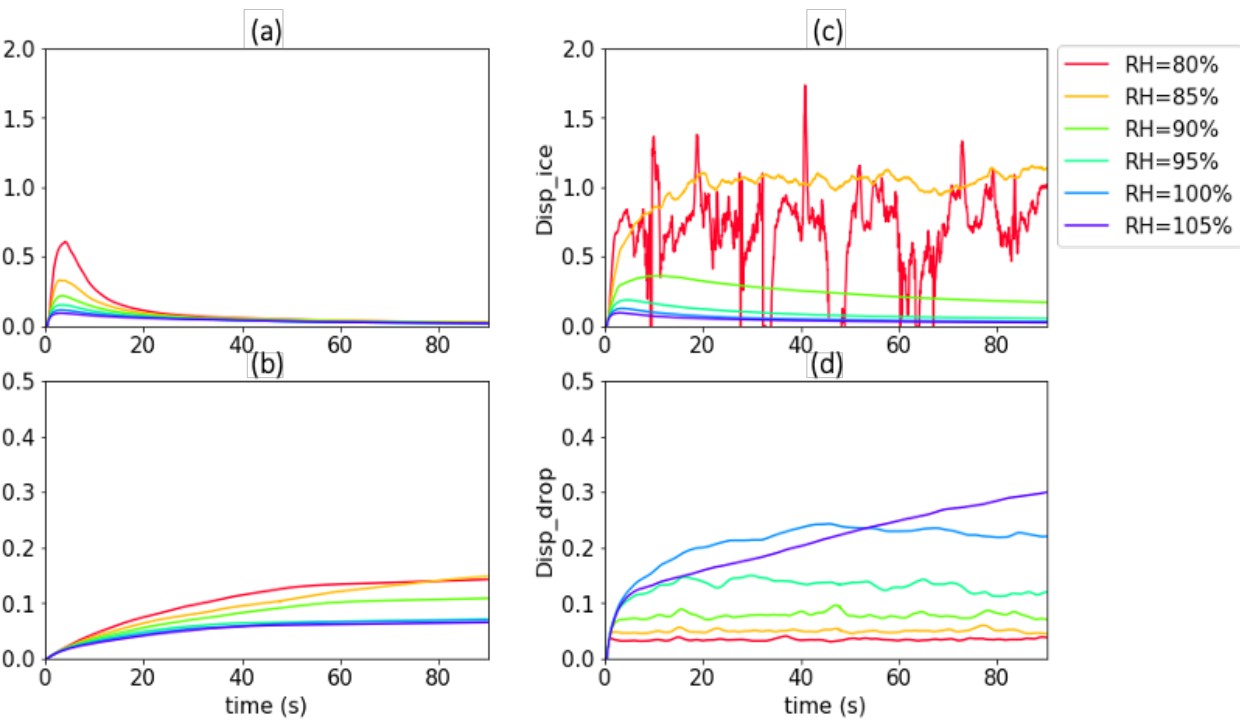


**Figure 5.** The relative dispersion of (top row) the ice size distribution and (bottom row) the droplet size distribution in air parcels initiated with different RH. The left column shows results from RH_GC, the right column from RH_noGC.

**4.2 Microphysical impact**

Next, we investigate the microphysical impact, such as the initial particle number concentration and particle size, on the ice
growth. In Group High_IN, the range of ice number concentrations tested is much higher than the typical number concentrations of natural ice observed in mixed-phase clouds, which usually is much lower than $1\ cm^{-3}$ in the atmosphere. However, it is well recognized that measuring the ice particles below $100\ \mu m$ is challenging and bears a high uncertainty. Ice particles with size $\ll 100\ \mu m$ can be largely under detected. This means that $1\ L^{-1}$ of ice particles $> 100\ \mu m$ could be a result




of ice crystal growth at a higher concentration. Therefore, the purpose of using this higher range is (1) to examine the sensitivity
of resulting large ice particle number concentration to the initial ice crystal number concentration and (2) to test the upper
bound effects of ice number concentration.

Changing ice concentration exerts a huge impact. At $N_i = 100 \ cm^{-3}$, ice particles quickly consume the water vapor (Figure
6 c, d) and droplets completely evaporate after 50 seconds. Therefore, the parcel is glaciated within one minute (Figure 6(e)).
Though both IWC and total water content (TWC) stay the highest of the three simulations, the mean ice radius could not grow
beyond 10 $\mu m$, as too many ice crystals or ice nucleating particles "overseeds" the cloud parcel and subsequently inhibits the
formation of precipitating-size particles. In contrast, with $N_i = 1 \ cm^{-3}$, the parcel maintains mixed-phase much longer than
the rest of the cases and sustains ice growth through the Bergeron process. Therefore, the formation of precipitating ice particles
and the long-term persistence of mixed-phase conditions requires a relatively low ice number concentration ($N_i \leq 1 \ cm^{-3}$)
and a high LWC environment.

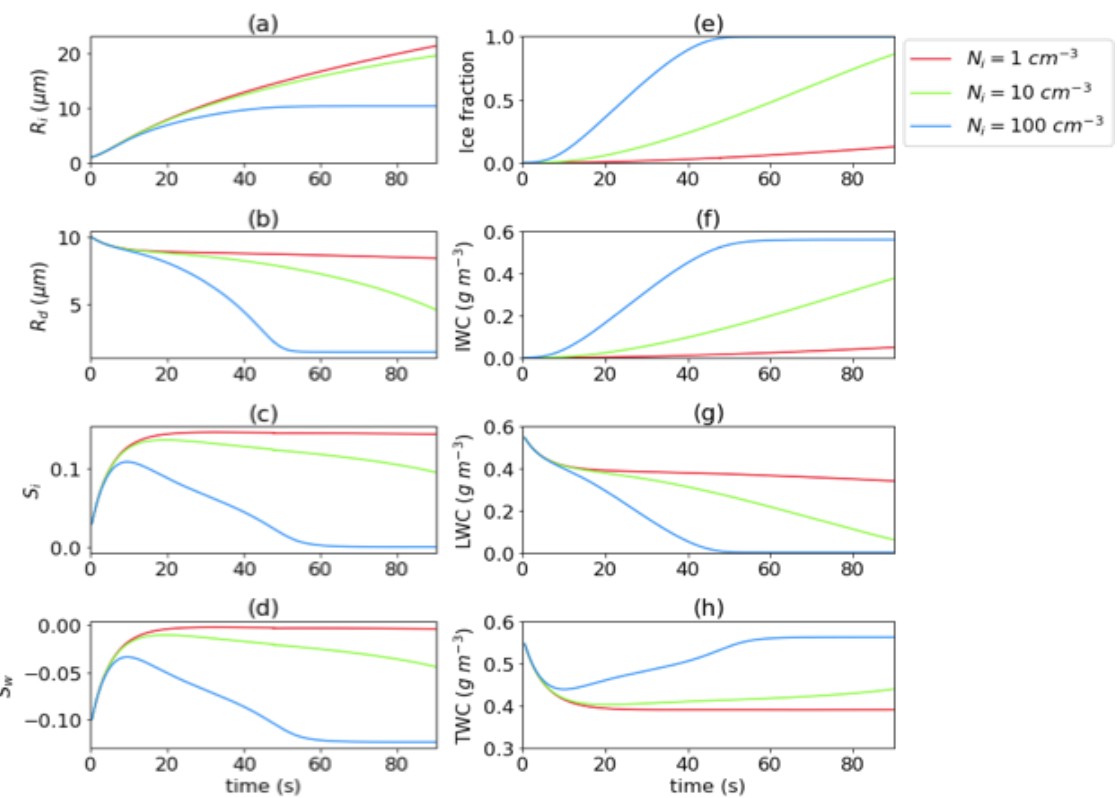

**Figure 6.** (a-d) are the same as Figure 2, but for Group High_IN. (e-h) are the corresponding evolution of ice fraction, IWC,
LWC, and total water content (TWC), respectively.


As is already demonstrated in High_IN above that for $N_i \leq 1 \ cm^{-3}$ the parcel favors effective ice growth in a high LWC
(0.55 $g \ cm^{-3}$). In Low_IN experiments, the simulations were initiated with a lower LWC environment by reducing its droplet
number concentration by a factor of ten (LWC = 0.055 $g \ cm^{-3}$, $N_d = 10 \ cm^{-3}$, $R_d = 10 \ \mu m$, see also Table 2) to examine
how ice number concentrations influence the ice growth in a low LWC. Unlike the High_IN, when $N_i \leq 1 \ cm^{-3}$ the growth
of ice and evaporation of droplets were insensitive to the ice number concentration (Figure 7a-b) compared to High_IN.



However, the higher number concentration can lead to a much higher glaciation rate (Figure 6e). The mean supersaturation only starts to diverge after droplets nearly evaporate around 40 sec (Figure 7c-d). And ice grows in general is slower than in High_IN shown in Figure 6a albeit that fewer ice particles are competing for water vapor. This implies even if the ice number concentration is low ($<< 1\ cm^{-3}$), a high LWC is necessary for rapid growth of big ice particles through diffusional growth.




**Figure 7.** Same as **Figure 6** but for Group Low_IN.

### 4.3 Turbulent mixing impact

To further examine how turbulent-mixing-induced supersaturation fluctuations affect the ice growth and the glaciation process,
we compare the results of the controlled experiment with the case without the turbulent mixing effects on the thermodynamic fluctuations. By comparing the two idealized simulations we can isolate the impact of turbulent-mixing on the microphysics at local scales ($\ll 1\ m$).

The results show that local thermodynamic fluctuations generated by the turbulent mixing cascaded from larger scales have a
negligible effect on the bulk (domain mean) properties as seen in Figure 8. The mean radius of the particles is almost identical except for a slightly smaller mean droplet radius in the thermodynamically-forced case. The mean radius is calculated based on the arithmetic average, it is found that the volume-weighted mean radius is not altered by the fluctuation (not shown), substantiated by the identical LWC in Figure 8 (g). As the small-scale fluctuations have a negligible effect on the bulk condensation, this implies that the microphysics parameterization on diffusional growth can neglect the effect from the sub-
grid-scale fluctuations on the droplet mean property.





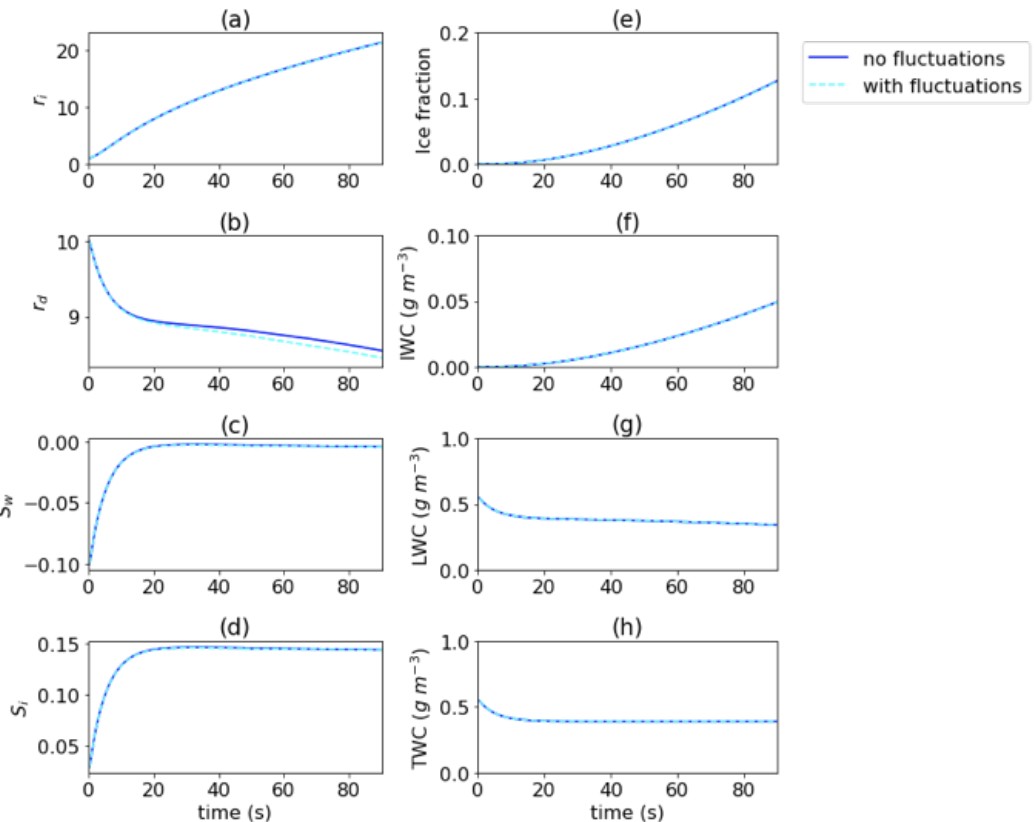

**Figure 8.** same as **Figure 6** but for Group Turb with (dashed line) and without (solid line) turbulent-mixing.

Albeit the same mean properties in both cases, the size distribution width broadening is significant. In our case, the particle distribution is monodispersed in the beginning. Therefore, the broadening is solely caused by the saturation fluctuations. We conducted another experiment without supersaturation fluctuations for comparison. It can be seen from the relative dispersion that no broadening happens in the case without supersaturation fluctuations (Figure 9) because particles are exposed to a fluctuated supersaturation environment.


   This supersaturation-fluctuation-induced broadening can be further substantiated by looking at spatial distribution of the supersaturation. Figure 10 displays the horizontal cross-section of the supersaturation fields at T=41 second in the RH_GC for RH=80%. On average, this case manifests an active Bergeron process, with a mean negative supersaturation with respect to water (~-0.3%) and mean positive supersaturation with respect to ice (~15.0%) (Figure 2c-d). This leads to effective ice growth

and droplet evaporation. However, Figure 10 shows turbulent mixing creates pockets of positive supersaturation with respect to water (with maximum value of 2.3% at this time point). This locally high supersaturation keeps the droplets from evaporation and even favors the droplet growth as shown in the PSD evolution in Figure 4(c). This effective broadening at a time scale of a few seconds within such a small domain again demonstrates the importance of small-scale fluctuations that could not be resolved by the commonly used cloud models or even large eddy simulations.






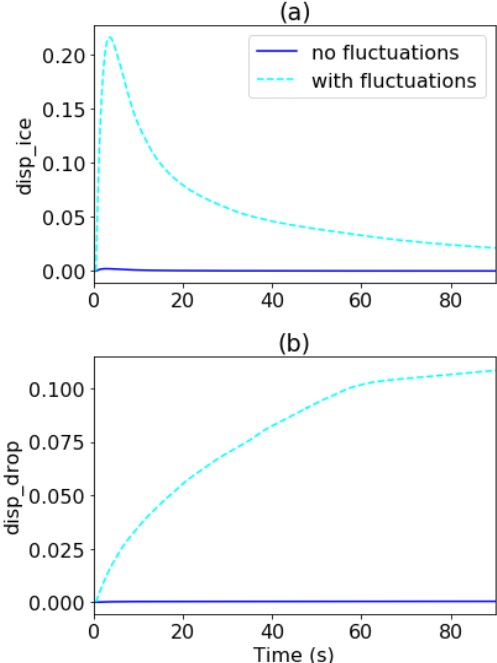

**Figure 9.** The relative dispersion of (a) ice and (b) droplets varies with time for cases with and without thermodynamic fluctuations in the temperature and mixing ratio fields, and thus no supersaturation fluctuations.







**Figure 10.** Horizontal cross section of the simulation at t = 41 second in the RH_GC group initialized with RH=80%. The filled color contour indicates the supersaturation with respect to water (%), and the contour lines refer to the supersaturation with respect to ice (%). The ice particles are marked as black dots, and droplets as orange dots. The size of the particles reflects the relative difference in the radii of droplets and ice particles.







## 5 Summary and discussion

In this study, we developed the first DNS model capable of simulating mixed-phase cloud microphysics at its native scales and presented the first results of such a model.

Six groups of experiments with initial conditions resembling the typical environments inside and outside of the cloud-top generating cells were conducted. By changing the environmental (macrophysical and turbulent) conditions and microphysical conditions, various potential factors affecting the ice growth are explored to unravel the favorable conditions for ice growth in GCs.

In summary, high LWC or high RH are critical to maintaining both effective ice growth and mixed-phase status. Generating cells with higher LWC and higher RH than the adjacent portion of the clouds provide ideal environments for efficient ice growth. In the meantime, solute effects from large aerosols can be important at the initial growth stage to maintain droplet growth and thus the increased LWC, this has an important implication for future cloud seeding operations to maintain high LWC for a more active Bergeron process.

Sensitivity studies on ice number concentrations show that the ice number concentration below $1\ cm^{-3}$, a typical range in the atmospheric clouds, a high LWC is needed for an efficient formation of big individual ice particles through diffusional growth. In addition, the ice radius growth rate is less sensitive to the ice number concentrations. However, a high ice number concentration $> 10\ cm^{-3}$ results in an early termination of the Bergeron process, fast glaciation of mixed-phase clouds, 385 suppression of the ice growth, which does not favor the formation of hydrometeors. This finding has an applicational implication on the winter-cloud seeding.

By comparing the cases with and without turbulent-mixing, it is found that small-scale thermodynamic fluctuations substantially broaden the particle size spectra. The effective broadening at a time scale as fast as a few seconds and at a length 390 scale down to cm-scales demonstrates the importance of small-scale fluctuations that could not be resolved by the commonly used cloud models or even large eddy simulations. However, the simulation also shows that the fluctuations have a negligible effect on the mean properties such as the mean radius and mean saturation ratio.

The current study represents the first attempt to use the DNS model to understand the mixed-phase cloud microphysics at local 395 scales. This first result reveals the capability of such an ultra-fine-resolution model to address the outstanding problems in mixed-phase cloud research. It bears some limitations and requires future work to fill the gaps. Firstly, studies that consider collisions between droplets and ice particles are needed to further investigate the turbulent effect on the evolution of particle size distribution through collisions. Secondly, the mixing of air parcels of different properties not only affects the fluctuations but also influences the mean properties. In addition to the turbulent mixing effects on thermodynamic fluctuations, impacts on 400 the mean temperature and mixing ratio also need to be considered to reflect the realistic entrainment-mixing process. Observations on the cooling rate, variation of mixing ratio at scales much larger than the DNS, and dynamical and microphysical information along the Lagrangian trajectory in a 3D high-resolution cloud model such as LES can help to better constrain the DNS for more realistic investigation on the mixed-phase processes at the microscale. Thirdly, the ice geometry information is important to determine its fall velocity and diffusional growth rate and needs to be considered to better represent 405 the evolution of ice particle size distribution (Korolev et al. 1999; Fukuta 1969; Avramov and Harrington 2010; Chen and Lamb 1994; Jensen et al. 2017; Jensen and Harrington 2015; Morrison et al. 2020). Ongoing modification is conducted to improve the model.





With the current model capability and ongoing development, the future application of the DNS on ice and mixed-phase
microphysics can also be promising in reproducing or comparing with the results in the laboratory experiments (Desai et al.
2019), which can provide process-level understanding of the experiments and help design new experiments and new laboratory
facilities with the well-validated model (Shaw et al. 2020).

**Acknowledgments**

This material is based upon work supported by the National Center for Atmospheric Research, which is a major facility
sponsored by the National Science Foundation under Cooperative Agreement No. 1852977. This work is also supported by
Idaho Power Company. We would like to thank Jeff French, Wojciech Grabowski, and Anders Jensen for their valuable
discussion. LX thanks Paul Field for the inspirational discussion on this topic that led to the development of the mixed-phase
DNS capability. Computing resources were provided by the Climate Simulation Laboratory at NCAR's Computational and
Information Systems Laboratory (CISL).

**Code and data availability**

The data of this study are available to download in Harvard Dataverse (Chen 2022). Figures were made

with matplotlib on the Jupyter Notebook. Part of the software associated with this manuscript for the

visualization is also stored in Harvard Dataverse (Chen 2022).

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
