# Peer review of "Mixed-phase Direct Numerical Simulation: Ice Growth in Cloud-Top Generating Cells"

_EGUsphere, 2022_

## Referee Comment (RC2)

**Review of "Mixed-phase Direct Numerical Simulation: Ice Growth in Cloud-Top Generating Cells"
by Chen et al. (egusphere-2022-1142)**

The manuscript describes the first direct numerical simulations of mixed-phase cloud microphysics. By conducting a large number of sensitivity studies, the authors analyze how the relative humidity and the liquid water content affect the Wegener-Bergeron-Findeisen process in a turbulent environment. They show that the results do not substantially deviate from idealized parcel simulations. The main differences are in the widths of the droplet and ice crystal size distributions.

While the results are not very exciting, this is an important contribution to the literature, as the interaction of turbulence and mixed-phase cloud microphysics is not very well studied. Thus, I support the manuscript's publication in Atmospheric Chemistry and Physics, subject to a few major and minor comments detailed below.

**Major Comments**

*Are K' and D' the same for ice and liquid particles?* The authors consider kinetic effects on the coefficients for thermal conductivity and water vapor diffusivity for ice crystal growth the same way they are used for liquid droplets. I have major concerns if this is acceptable (see, e.g., Zhang and Herrington 2014).

*The dry aerosol size seems to be unrealistically large.* A droplet growing from a dry aerosol of 1 μm in radius needs to be considered a giant nucleus. Initializing all liquid droplets with such a large dry aerosol is not realistic, as only a minority of cloud droplets are grown on aerosol particles as large as this.

*Connect results to theory.* There is a lot of theory on the Wegener-Bergeron-Findeisen process in the literature. As this study builds upon that work and extends it by including turbulence, it would be worthwhile to include some of the calculations of, e.g., Korolev and Field (2008).

**Minor Comments**

L. 13: While I see this is the abstract, please explain briefly what cloud-top generating cells are. I do not consider them common knowledge.

Ll. 23 – 33: Small-scale mixed-phase cloud processes are not only a sub-grid scale problem. The lack of understanding addressed in this study is much more fundamental. Thus, the authors should not understate the physics investigated here.

Ll. 63 – 64: This applies to all mesoscale models, not only WRF-type models. Moreover, WRF is not defined.

Ll. 87 – 88: If the effects of T' and qv' have not been considered before, was the spectral broadening shown in previous studies only due to temperature fluctuations caused by fluctuations in the vertical velocity?

Ll. 90 – 91: What is "the early stage of a mixed-phase cloud"?

Ll. 104 – 106: The crystal habit should depend on temperature and humidity.

Eqns. 4 and 5: Are those \delta the same as \sigma?

Tab. 2: I hope this is a typo, but a dissipation rate of 10 m^2 s^-3 is about 10 000 times larger than what one would expect in a cloud with the LWCs used here.

Ll. 195 – 199: This is not a steady state. It only looks like a steady state because the simulation time is too short to detect any changes. Clarify this.

Ll. 200 – 203: How are these statements supported? To which figure are the authors referring to?

Ll. 267 – 268: It is quite interesting that the ice crystal size distribution is only susceptible to turbulence when the environment is almost ice-saturated. This finding should be highlighted in the

conclusions/summary, as it might help to constrain the conditions in which those interactions are important.

Fig. 9: Could you speculate on what would happen to the relative dispersions if the supersaturation fluctuations are turned off at, e.g., t = 20 s? Would a non-zero relative dispersion be maintained, or would it decrease to zero?

Ll. 346 – 354: This analysis is entirely qualitative. Is it possible to quantify the correlations between liquid droplets, ice crystals, and the supersaturation?

**Technical Comments**

General: Please increase the line spacing in the draft version of the manuscript. The draft is very hard to read.

Ll. 18 ff.: State units in upright characters.

Ll. 46 – 48: There is something wrong with the sentence starting with "A majority […]".

Ll. 92 – 93: Double reference to Pruppacher and Klett (2010).

L. 102: Add units to Rv.

Tab. 1: I suggest replacing liters (L) with cubic centimeters (cm^3). Last row: No unit for ice concentrations.

All figures: Add the simulation group name to the title of each panel.

Fig. 3, caption: This should be an LWC of 0.002 g m^-3.

Fig. 4: Increase the quality of the plot. The panels are blurry.

Fig. 5a and b: Ordinate labels are missing.

**References**

Korolev, A., & Field, P. R. (2008). The effect of dynamics on mixed-phase clouds: Theoretical considerations. *Journal of the Atmospheric Sciences*, *65*(1), 66-86.

Zhang, C., & Harrington, J. Y. (2014). Including surface kinetic effects in simple models of ice vapor diffusion. *Journal of the Atmospheric Sciences*, *71*(1), 372-390.

---

## Author Comment (AC1)

Response letter for ACP2022

**Review #1**

We thank the reviewer's careful review and valuable feedback. We have carefully considered each of them and made appropriate revisions to the manuscript. Below are the point-by-point responses.

1) I would like to know more details about the simulations, such as grid spacing, time stepping and used numerics. It would be also interesting to know if the droplets/ice sediment, and with which velocity.

In our simulations, dx=0.15625m and dt=0.472×10^(-4) s in order to adequately resolve the turbulent eddies and particle diffusional growth. The domain size is 20cmx20cmx20cm. Droplets/ice fall due to gravity, but we apply a triply-periodic boundary so that particles falling out of the bottom will re-enter from the top to assure a steady number of particles for statistical analysis. Therefore, no sedimentation impact is considered in this study. Since we focused on the initial ice growth, and all droplets and ice are smaller than 25 microns, sedimentation impact can be mild, if not insignificant, to affect the particle size distribution. However, for later stages in which particles exceeding 50 microns, sedimentation can be crucial.

We added the following descriptions in Sections 2 and 3 to give more details of the simulation and model:

Line 79-81: "In this modeling framework, droplets are modeled as Lagrangian particles and are treated as inertial particles, subject to both turbulent flow and gravity. The simulation domain is defined by a triply-periodic boundary condition. "

Line 181-186: "To accurately capture all turbulent scales (i.e., scales down to Kolmogorov length scales) in the computational domain, the spatial resolution was set to 0.15625 m. The time step was set to be very small ($\Delta T=0.472×10^{-4}$ s) to ensure that the smallest eddies and the particle relaxation timescales could be resolved. The droplet relaxation time scale defines how fast a droplet's surface temperature adjusts to its steady-state value (see also the discussion of relaxation timescales in Vaillancourt et al. 2001). The domain size was 20 cm x 20 cm x 20 cm, and with a droplet number concentration of 100 cm^(-3), 800,000 droplets were simulated and tracked within the domain. "

2) In the description of the measured environmental conditions (table 1) it would be useful to have the mixing ratios qc und qi and the number concentrations qnc and qni. It is also said that drizzle was measured, but I guess this has no influence on the experiments.

In this dataset, only particles exceeding 100 microns were available. Therefore, we do not have measurements on the number concentration of small droplets and small ice particles.

3) The DNS experiments have a dissipation rate of 10m2s-3. Is this not too high for such a cloud?

Apologize for the typo. The dissipation rate is actually $10 \; cm^2 s^{-3}$, and is of comparable magnitude to the flight measurements.

4) Please state the mean temperature in the experiments.

We set the mean temperature initially to 259.53 K, and it evolves with time due to droplet evaporation and ice deposition. The equilibrium value varies with RH (see figure below).

[Figure]

Fig. A: The evolution of temperature and water vapor mixing ratio for RH_GC case.

5) Why do you use a standard ice concentration in the DNS of 1000 l-1 when only 1l-1 was measured?

We justified this setup in the manuscript: "In Group High_IN, the range of ice number concentrations tested is much higher than the typical number concentrations of natural ice observed in mixed-phase clouds, which usually is much lower than $1 \; cm^{-3}$1 in the atmosphere. However, it is well recognized that measuring the ice particles below 100 µm is challenging and bears a high uncertainty. Ice particles with size <<100 µm can be largely under detected. This means that $1 \; L^{-1}$ of ice particles > 100 µm could be a result of ice crystal growth at a higher concentration. Therefore, the purpose of using this higher range is (1) to examine the sensitivity of resulting large ice particle number concentration to the initial ice crystal number concentration and (2) to test the upper bound effects of ice number concentration. "

In addition, the domain size is 20cmx20cmx20cm. 1 $L^{-1}$ corresponds to only 8 ice crystals in the entire domain. To get a statistically robust size distribution and statistics, it is also beneficial to use a higher number of ice crystals. This limitation can be overcomed by using a larger domain size, however the computational cost is much higher.

6) The experiment RH_noGC starts with unusually low radius of droplets (R=1.5 µm), which I do not think can be easily found in clouds. It is then not surprising that all liquid water quickly evaporates if RH<100%. Please motivate these experiments.

The main purpose of running experiments with a low LWC environment (i.e. small droplets with same number concentration as that of RH_GC) is to demonstrate the importance of the presence of large drops to ice growth for the same RH environments and the sensitivity of ice growth to RH at different LWC environments.

We added "The primary objective of conducting experiments in low LWC environments is to demonstrate the crucial role of large drops in ice growth under the same RH conditions, as well as the susceptibility of ice growth to changes in RH at varying LWC environments. " to Line 166-169.

In addition, it is also interesting to see in Fig.2 that for droplets with a thin wet layer (R=1.5 µm in RH_noGC), the solute effect is strong that the Begeron process is suppressed at the beginning and droplets grow for the first few seconds when RH>90%. In RH_GC with a larger droplet radius (R=10 µm), droplets grow only when RH>100%. (See detailed discussion in Line 240-258)

7) It is a common assumption in weather models/LES with a time step 1-20 seconds that liquid water is in thermodynamic equilibrium, but not the ice. The underlying assumption is that water condensation/evaporation is much faster than ice processes. Could the authors comment on how good this approximation is? These experiments could be very useful to verify the above-described approximation.

We thank the reviewer for raising this very interesting question. Given the experiment results, we found that in a mixed-phase environment, it took ~20-30 seconds for the vapor field (saturation ratio) to reach steady-state, and this relaxation time scale depends on both LWC and RH. Our studies show neither the droplets nor the ice reaches thermodynamic equilibrium in this closed system for the length of simulation (T=90s), however, their growth rate and the size spectrum width (Fig. 5) reach steady-state at the same timescale.

The simplified assumption that water condensation/evaporation is faster than ice processes is based on the single phase cloud (i.e. warm cloud or ice cloud) that the droplets or ice react to moisture fields independent of one another. The phase relaxation time (which defines the time taken for saturation ratio relaxes exponentially to 100%)

$\tau_{phase} = (4\pi\kappa N\bar{R})^{-1}$, $\kappa = 2.55 \times 10^{-5} m^2 s^{-1}$ is the diffusivity of water vapor. For droplet with $N = 100\ cm^{-3}$ and $\bar{R} = 10\ \mu m$, $\tau_{phase} = 3.12\ s$, for ice with much lower number concentration ($N = 1\ cm^{-3}$ and $\bar{R} = 1\ \mu m$), $\tau_{phase} = 3121\ s$. However, this assumption does not hold for an evolving microphysics (i.e. number concentration and mean radius changing over time), and a mixed-phase environment (constant water vapor exchange between ice and water via the Bergeron process). For a mixed-phase condition, ice, droplets, and thermodynamic fields interact in a concurrent manner, and both particles react to the thermodynamic fields in the same phase-relaxation timescale that evolves with time in the system. Both particles react to the thermodynamic fields in the same phase-relaxation timescale that evolves with time in the system, resulting in a much slower reaction timescale to reach steady-state than the phase relaxation time scale defines.

We therefore added the above discussion to Line 199-211:

"In weather models or LES, a time step of 1-20 seconds is typically used with the assumption that liquid water is in thermodynamic equilibrium, but not for the ice. This assumption is based on the approximation that the phase relaxation time (the time it takes for the saturation ratio to relax exponentially to 100%) for droplets is faster than that of ice. This simplified assumption that water condensation/evaporation is faster than ice processes is based on the single-phase cloud (i.e., warm cloud or ice cloud) where droplets or ice respond to moisture fields independently of one another. The phase relaxation time ($\tau_{phase}$) can be calculated as ($\tau_{phase} = (4\pi\kappa N\bar{R})^{-1}$), where $\kappa$ is the diffusivity of water vapor ($\kappa = 2.55 \times 10^{-5} m^2 s^{-1}$) and and $\bar{R}$ is the mean droplet radius. For a liquid cloud with a number concentration of $N = 100\ cm^{-3}$, and mean radius of $\bar{R} = 10\ \mu m$, $\tau_{phase} = 3.12\ s$; for an ice cloud with a much lower number concentration (N=1 $cm^{-3}$ and $\bar{R} = 1\ \mu m$ in our case), $\tau_{phase} = 3121$ s. However, this assumption does not hold true for evolving microphysics (i.e., number concentration and mean radius changing over time) or in mixed-phase environments where water vapor exchange occurs between ice and water drops. In mixed-phase conditions, ice, droplets, and thermodynamic fields interact concurrently. Both particles react to the thermodynamic fields in the same phase-relaxation timescale that evolves with time in the system, resulting in a slower reaction timescale to reach steady-state than the one defined by the droplet phase relaxation time scale. "

 And added to Line 384-389:

"The results show that in high LWC conditions, the saturation field reaches a thermodynamic steady-state within 20 to 30 seconds. This steady-state is maintained through an unstable microphysical condition, in which the loss of water vapor from ice growth is continuously replenished through the evaporation of droplets (i.e., the WBF

process), leading to a steady-state RH. The timescale for reaching steady-state in this mixed-phase environment is much slower than in a pure liquid environment. As a result, in a mixed-phase condition, where ice, droplets, and thermodynamic fields interact concurrently, the phase-relaxation timescale is not a reliable measure of the thermodynamic response time. "

**Review #2**

We thank the reviewer's careful review and valuable feedback.  We have carefully considered each of them and made appropriate revisions to the manuscript. Below are the point-by-point responses.

**Major Comments**

Are K' and D' the same for ice and liquid particles? The authors consider kinetic effects on the coefficients for thermal conductivity and water vapor diffusivity for ice crystal growth the same way they are used for liquid droplets. I have major concerns if this is acceptable (see, e.g., Zhang and Herrington 2014). The dry aerosol size seems to be unrealistically large. A droplet growing from a dry aerosol of 1 µm in radius needs to be considered a giant nucleus. Initializing all liquid droplets with such a large dry aerosol is not realistic, as only a minority of cloud droplets are grown on aerosol particles as large as this. Connect results to theory. There is a lot of theory on the Wegener-Bergeron-Findeisen process in the literature. As this study builds upon that work and extends it by including turbulence, it would be worthwhile to include some of the calculations of, e.g., Korolev and Field (2008).

We appreciate the reviewer's concern. The different kinetic effect on ice vapor diffusion described in Zhang and Herrington (2014) is mostly attributed to the shape of ice crystal, as vapor diffusion rate onto the ice surface can be very different if ice takes non-spherical shape. In our study, we focus on the very small ice particle (<<100 microns) and therefore ice crystals are assumed to be spherical and are treated in the same way as droplets. However, we acknowledge that this is not a very accurate assumption. We therefore mentioned this limitation and suggestions for distinguishing kinetic effects between ice and droplets for future studies.

Line 100-102: "Note that the ice crystals are spherical in this study and therefore the kinetic effects of the diffusivity and conductivity on ice crystals are treated in the same way as water. Zhang and Herrington (2014) demonstrated that when ice crystals take non-spherical shapes, the kinetic effects can be very different, and their shape parameter needs to be considered. "

We set dry aerosol size as 1 micron, which indeed is larger than typical aerosol size. We use this large initial size to examine the impact of the solute effects of droplets in the ice growth (discussion in Line 240-258). It is interesting to see that the presence of giant dry aerosol size (RH_noGC group) can slow down the WBF process by exerting strong solute effect to maintain droplet growth in a longer timescale than in the case where the aerosol solute is dissolved in the

droplets (RH_GC). In the revision, we have addressed this limitation in the manuscript and justified as follows:

Line 158-160: "The giant dry radius of 1 μm is not a typical aerosol radius, however it is used in this experimental setup for the purpose of examining the upper bound impact of the aerosols's hygroscopic effects on the ice growth and the WBF process. "

We also add some discussions of the theoretical studies by Korolev (2008) to connect with our studies: "The theoretical study from Korolev (2008) found that for a typical LWC and IWC, ice growth is significantly less sensitive to the vertical velocity in mixed-phase clouds than the droplet growth. This finding is consistent with our finding, as the vertical velocity determines the saturation ratio (i.e., RH). " (Line 278 - 280)

**Minor Comments**

L. 13: While I see this is the abstract, please explain briefly what cloud-top generating cells are. I do not consider them common knowledge.

Thanks for this suggestion. We have added the definition of generating cells in the abstract: "This paper examines the conditions that favor effective ice growth in the cloud top generating cells (GCs), which are small regions of enhanced radar reflectivity near cloud tops. GCs are commonly observed in many types of mixed-phase clouds and play a critical role in producing precipitation in rain or snow. "

Ll. 23 – 33: Small-scale mixed-phase cloud processes are not only a sub-grid scale problem. The lack of understanding addressed in this study is much more fundamental. Thus, the authors should not understate the physics investigated here.
We highly appreciate the reviewer's comment. It is indeed a more fundamental problem than the solely numerical issue, therefore, we modified the statement as follows:

"It is well-recognized that the model representation of natural ice is not accurate due to a poor understanding of the physics related to, and therefore the poor model representation of, the ice and mixed-phase microphysical processes such as the ice nucleation mechanisms, representation of supercooled liquid water, and the interactions between the ice-liquid phase hydrometeors (Korolev and Milbrandt 2022; Korolev et al. 2017; Koop and Mahowald 2013; Ovchinnikov et al. 2014). "

Ll. 63 – 64: This applies to all mesoscale models, not only WRF-type models. Moreover, WRF is not defined.
We have removed "WRF" and modified the sentence to "The structure of the flow at this length-scales is difficult to measure using in-situ data and is also unable to resolve in all mesoscale models. "

Ll. 87 – 88: If the effects of T' and qv' have not been considered before, was the spectral broadening shown in previous studies only due to temperature fluctuations caused by fluctuations in the vertical velocity?

The reviewer is correct. The spectral broadening shown in previous studies was mainly caused by the vertical velocity, and the supersaturation fluctuation caused solely by vertical velocity fluctuations without external thermodynamic forcing are very small (see Figure 4 in Chen et al. 2021)

Ll. 90 – 91: What is "the early stage of a mixed-phase cloud"?

Early stage refers to the initial ice growth stage. However, we realized it is not an accurate term, therefore we removed it from the sentence: "The purpose of this model is to simulate the initial ice growth (r<100 μm) in mixed-phase clouds…"

Ll. 104 – 106: The crystal habit should depend on temperature and humidity. Eqns. 4 and 5: Are those \delta the same as \sigma? Tab. 2: I hope this is a typo, but a dissipation rate of 10 m^2 s^-3 is about 10 000 times larger than what one would expect in a cloud with the LWCs used here.

Thank you for the correction. We have corrected the sentence.
The \delta in equations 4 and 5 are the same as \sigma. We replaced them with \sigma to reduce confusion.
Tab. 2: Thanks for pointing out the typo (missing "c"). The unit should be cm^2/s^-3. We have corrected it in the revision.

Ll. 195 – 199: This is not a steady state. It only looks like a steady state because the simulation time is too short to detect any changes. Clarify this.

We agree with the reviewer that it is a true steady-state and only remains stable for the timescale we are studying. To reduce confusion, we modified it to:
"Figure 2 shows the response of the mean particle radius and mean supersaturation to various initial RH. In the RH_GC group, the mean supersaturation adjusts to a quasi-steady state within 20-30 seconds and remains relatively unchanging for the duration of the simulation. "
Accordingly, we have changed all "steady state" to "quasi-steady state"

Ll. 200 – 203: How are these statements supported? To which figure are the authors referring to?

We added the corresponding figure numbers to supporting the statements:
"The RH_GC experiments demonstrate that the saturation at its quasi-steady state is insensitive to the initial RH when at high LWC conditions (Figure 2c-d). During the quasi-steady state, all ice grows at a relatively constant rate and is less sensitive to the initial relative humidity compared to droplet growth (Figure 2a). "

Ll. 267 – 268: It is quite interesting that the ice crystal size distribution is only susceptible to turbulence when the environment is almost ice-saturated. This finding should be highlighted in the conclusions/summary, as it might help to constrain the conditions in which those interactions are important.

We agree that the turbulence-induced broadening of ISD is mostly prominent when the environment is almost ice-saturated. It should be noted that the broadening is also efficient in the initial time when in supersaturated environments. E.g., Fig 4 (d) ISDw for RH>=95% at the first 20 seconds, although it maintained a relatively constant width afterwards.
To highlight this finding, we added the following statement in the conclusions:
" It is found that droplet spectral broadening is in general more sensitive to turbulent fluctuations; ice spectral width remains very similar when it is supersaturated with respect to ice, and broadening of ice size distribution is mostly prominent when the environment is close to ice saturation. "

Fig. 9: Could you speculate on what would happen to the relative dispersions if the supersaturation fluctuations are turned off at, e.g., t = 20 s? Would a non-zero relative dispersion be maintained, or would it decrease to zero?

We thank the reviewer for raising this interesting discussion.

If the fluctuations in supersaturation are turned off in the middle of the simulation, given that the particle sizes are different, their radii would grow or shrink at a rate approximately proportional to S/r (or dr^2/dt ~ S), where S is the supersaturation ratio, and r is the radius of the particle.

As the parcel is supersaturated with respect to ice (S_i>0), therefore, it is deduced that larger particles grow slower than the smaller particles, leading to a decrease in the relative dispersion. For droplets, the supersaturation ratio is negative (S_w<0). Therefore, small droplets also evaporate faster than larger droplets, leading to a more dispersed size spectral.

So we speculate that the ice spectral will be narrowed while the droplet spectral widens. This probably can explain why the particle distribution tends to be wider in a subsaturated (or low RH) environment such as in Fig.4c.

Ll. 346 – 354: This analysis is entirely qualitative. Is it possible to quantify the correlations between liquid droplets, ice crystals, and the supersaturation?

We thank the reviewer's suggestion and added a more quantitative evaluation of the correlation between liquid, ice, and the supersaturation field (Fig. B below, also Figure. 11 in the manuscript).

We added the following analysis in the manuscript between Line 352-360:

" To quantitatively evaluate the spatial correlation between the droplet, ice, and supersaturation, Figure 11 (a-b) shows the Pearson correlation between the particle radius and the supersaturation value at the particle's location (i.e., the Lagrangian supersaturation). For droplets, the correlation between the size and supersaturation field is always positive (with p-value < 0.001). Correspondingly, the supersaturation fluctuations are strong in the first 10-20 seconds, leading to the strongest broadening. For ice particles, the correlation is positive before around 10 seconds (with p-value < 0.001) and becomes insignificant afterwards. Therefore, compared to ice particles, droplet spectral broadening is more sensitive to the supersaturation fluctuation fields. This can also be substantiated by the evolution of size distributions in Fig. 4 (c-d): the ice spectral broadening due to supersaturation fluctuations is mostly significant in the first 10 seconds while the droplet broadening is notable for the first 50 seconds. "

[Figure]

Fig. B: (Upper two panel) The Pearson correlation coefficient (a) between the droplet size and the Lagrangian supersaturation and (b) between the ice radius and the Lagrangian supersaturation for the first 50 seconds in Group RH_GC. The pale-colored dashed lines indicate p-value of the corresponding correlation coefficient. (c) The standard deviation of supersaturation fluctuation for the same period as in (a-b).

Technical Comments General: Please increase the line spacing in the draft version of the manuscript. The draft is very hard to read.

We apologize for the inconvenience. We have increased the line spacing in the revision.

Ll. 18 ff.: State units in upright characters.
Its original format of cm^-3 in the manuscript is in upright characters.

Ll. 46 – 48: There is something wrong with the sentence starting with "A majority […]".

We apologize for the disordered sentence. It has been modified to the following:

"The bulk of the research has centered on warm-phase clouds (e.g., Ayala et al. 2008; Grabowski and Wang 2013; Li et al. 2020; Gotoh et al. 2016, 2021; Saito et al. 2019; Chen et al. 2018a, b, 2020, 2021) with a limited number of studies investigating ice-phase clouds (Vowinckel et al. 2019a, b). In contrast, mixed-phase processes have received very little attention to date, despite their importance and complexity."

Ll. 92 – 93: Double reference to Pruppacher and Klett (2010).
We have removed the duplication.

L. 102: Add units to Rv. Tab. 1: I suggest replacing liters (L) with cubic centimeters (cm^3). Last row: No unit for ice concentrations. All figures: Add the simulation group name to the title of each panel. Fig. 3, caption: This should be an LWC of 0.002 g m^-3. Fig. 4: Increase the quality of the plot. The panels are blurry. Fig. 5a and b: Ordinate labels are missing.

Added the unit to Rv
Tab. 1: Changed all L^-1 to cm^-3 and added the unit to the last row.
Added simulation group names to the title of each panel in all figures
Fig. 3: corrected the caption.
Fig.4: we replotted the figure with better quality.
Fig.5a-b, added labels

References
Korolev, A., & Field, P. R. (2008). The effect of dynamics on mixed-phase clouds: Theoretical considerations. Journal of the Atmospheric Sciences, 65(1), 66-86.

Zhang, C., & Harrington, J. Y. (2014). Including surface kinetic effects in simple models of ice vapor diffusion. Journal of the Atmospheric Sciences, 71(1), 372-390.

Chen, S., Xue, L., & Yau, M. K. (2021). Hygroscopic seeding effects of giant aerosol particles simulated by the Lagrangian-particle-based direct numerical simulation. Geophysical Research Letters, 48, e2021GL094621. https://doi.org/10.1029/2021GL094621

---

## Referee Report (RR1)

**Review of "Mixed-phase Direct Numerical Simulation: Ice Growth in Cloud-Top Generating Cells" by Chen et al. (egusphere-2022-1142)**

My comments have been considered with great care. I now support the publication of the manuscript. I have a few technical comments below that the authors might like to consider. Line numbers refer to the tracked-changes version of the manuscript.

**Technical Comments**

Consider writing about "quasi-stationarity" instead of "quasi-steady state", as the prior seems to be more correct in my eyes.

L. 16: "precipitation from rain or snow", not "precipitation in rain or snow".

Ll. 107 – 110: "Note that ice crystals are assumed to be spherical in this study", not "Noted that the ice crystals are spherical in this study".

L. 191: I believe that the spatial resolution of 0.15625 m is a typo. It should be much smaller to resolve the Kolmogorov lengthscale.

L. 191: Use a lower-case "t" for the time step.

Ll. 209 – 214: Introduce $\tau_{phase}$ the first time it is mentioned (l. 209). The equation for $\tau_{phase}$ lacks an "N" (l. 213). The diffusivity of water is usually represented by a "D" or "$D_v$" (l. 213), a "K" is usually used for thermal conductivities.

Fig. 11a and b: Name the panels "droplet-supersaturation correlation" and "ice-supersaturation correlation", not just "droplet" or "ice".